# Contrast-enhanced Micro-CT imaging of a foetal female pelvic floor reveals anatomical details

Dea Aaldijk[1]*, Sebastian Halm[1], Diana N. Liashchenko[2], David Haberthür[1]

**1** Institute of Anatomy, University of Bern, Bern, Switzerland, **2** Head of the Human Anatomy Department, Orenburg State Medical University (Russia), Orenburg, Orenburg Oblast, Russia

* dea.aaldijk@unibe.ch

## Abstract

### Background

The pelvic floor is a highly important structure for the stability of the pelvis, providing support for the organs that lie within it. Until today, the detailed anatomy of the female perineal centre and the exact course of surrounding muscles remain controversial. We demonstrate a method to non-destructively obtain high-resolution contrast-enhanced x-ray microtomography images from a long-fixed sample and thereby aim to contribute to the detailed anatomical knowledge about the female pelvic floor.

### Materials and methods

A human foetal pelvis of 20–21 weeks gestational age, formalin-fixed for 4 years, was immersed in Lugol solution and tomographically scanned periodically to document the staining process. The influence of the former fixation time was addressed by comparison with a short time fixed mouse pelvis. High-resolution imaging was performed using μCT, with detailed anatomical analysis supported by segmentation and 3D reconstruction.

### Results

Lugol staining of long-fixed tissue was effective and showed no disadvantages compared to short-fixed tissue. Lugol staining and high-resolution μCT images provided a nicely stained image-stack with clearly identifiable tissue types. The anatomy of the foetal pelvis and its structures were resolved in detail. Interconnections between the external anal sphincter, the bulbospongiosus muscle and the superficial transverse perineal muscle could be shown within the perineal centre. There was no evidence for a skeletal muscle that corresponded to the formerly described deep transverse perineal muscle, instead there was cloudy-looking tissue, most likely smooth muscle fibres dispersed in connective tissue in a 3-D arrangement.

**Data availability statement:** All relevant data are within the paper and its Supporting Information files. As stated in the paper, the full high-resolution tomographic dataset of the foetal sample at the last staining timepoint is available online for viewing at https://web-knossos.org/datasets/Institute_of_Anatomy/Foetus02_Lugol_15pct_152d_rec/view

**Funding:** The author(s) received no specific funding for this work.

**Competing interests:** The authors have declared that no competing interests exist.

## Conclusions

X-ray microtomography of Lugol-stained tissue is an excellent method to gain anatomical details in high resolution, in a non-invasive and non-destructive way, independently of the fixation time. Using this method, the topographical relationships of the pelvic floor muscles could be illustrated, showing their linkage within the perineal centre.

## Introduction

The female pelvic floor is a crucial anatomical structure for the stability and health of the lesser pelvis and, consequently, the organs that it contains. Many gynaecological pathologies arise due to the lack of support from the pelvic floor, such as prolapse of the uterus, bladder, or rectum, as well as incontinence. These conditions are widespread among elderly women. The prevalence of at least one pelvic floor disorder (urinary incontinence, faecal incontinence, or pelvic organ prolapse (POP)) in US women was estimated at 23.7%, with numbers increasing with age, parity and weight [1]. Thus, nearly a quarter of the female population is thought to be affected by these conditions. The prevalence of symptomatic POP in females was estimated to be between 8.3% and 19% [2–4]. A recent study identified a significant risk for POP in nulliparous women [5]. Training of the pelvic floor muscles has been shown to improve symptoms and decrease the severity of POP [6,7], highlighting the importance of muscular stability in a healthy pelvic floor. Thus, to fully understand the pelvic floor, its associated diseases, and their therapies, detailed knowledge of its healthy anatomy is paramount.

A large body of anatomical studies has described the female pelvic floor in the last decades, albeit with contradictory results. To date, there is no consensus on three important aspects:

1. The existence of the deep transverse perineal muscle (DTP) as a part of the perineal membrane (PM) in females is controversial. Fritsch et al. stated that the DTP does not exist in females, explaining this by the fact that – unlike in males – the urethra does not require securing to the bones by transverse fibres of the external urethral sphincter since it terminates at this level [8]. Some authors even questioned the existence of the DTP in males [9]. In contrast, other studies confirmed the existence of the DTP in females, though without attachment to the pubic bone [10,11]. In an ultrasound study from Santoro et al., the existence of the DTP was doubtful and therefore it was excluded from further analysis [12], whereas Plochocki et al. found the DTP only in 16.7% of examined specimens [13] and Guo et al. identified it in only 1 out of 22 pelvic MRI scans [14].

2. Some studies stated that the muscles from the pelvic floor attach to the perineal body (PB), while others postulated that these muscles cross uninterruptedly to the other side, forming a sling. A connection of the bulbospongiosus muscle (BS) with the external anal sphincter muscle (EAS) of the opposite side has been described

in various studies [13,15–18]. Shafik et al. argued that the muscles of the perineal region function as two-bellied muscles, rather than them ending in the PB. In gross dissection, they not only observed the continuity of the EAS with the BS on the other side, but also the continuity of the superficial transverse perineal muscle (STP) from both sides [15]. A recent study by Baramee et al. confirmed this crossing of fibres via gross dissection of the female perineal region [19]. The same pattern was found in a study by Zifan et al., but without specifying whether they were referring to the superficial or deep transverse perineal muscle. They used fibre tracking in magnetic resonance imaging (MRI) to visualise the course of the muscles [20]. Additionally, they also performed an X-ray microtomography (µCT) scan of the PB, showing the crossing of the fibres, though this was only done on a male subject, so the documentation in the female body remained missing. On the other hand, gross dissection of male and female cadavers revealed the continuity of the BS with the superficial portion of the EAS in a study by Plochocki et al., but denied the crossing to the muscle belly of the opposite side [13]. Larson et al. also rejected the idea of crossing fibres, but described their insertion into the perineal body instead, seen in a 3D analysis of thin-slice MRI [21]. This view was partly confirmed by Santoro et al., who analysed the PB with endovaginal ultrasound and gross dissection. They found the bulbospongiosus and the STP to insert into the caudal part of the PB (without crossing) and the EAS to pass through the inferior margin of it [12]. According to Wu et al., the PB solely serves as the attachment site for the pelvic floor muscles [10].

3. For a long time, the term "perineal body" was used to describe a mass of tendinous tissue in the centre of the pelvic floor, where surrounding muscles attach [22,23]. Recent publications suggested it to be seen rather as an area than an anatomical entity, emphasising the continuity of muscular structures in the pelvic outlet area [21,24,25]. Numerous descriptions of connections between the levator ani (LA) muscles and the perineal muscles have been published, providing evidence that this area should rather be viewed as an interacting complex instead of singular muscle entities [24]. A more recent view of this complex is the three-muscle sling theory, which describes an anterior, middle and posterior sling that all contribute to pelvic organ support [18,19,24].

Reasons for the ongoing ambiguity surrounding pelvic floor anatomy may include the methods used. Gross dissection remains a widely used technique that can give important insights in anatomical studies, but the expectations and experience of the performing anatomist may influence the outcome of the dissection. Additionally, detaching pelvic floor structures from their bony insertions could alter their topographic relationships [26,27]. Other studies have used MRI as a non-destructive method to study anatomical structures. Unfortunately, the resolution in MRI images with voxel sizes in the millimeter range is often not enough to resolve the structures of interest [11,28–30]. Parikh et al. created a three-dimensional model of a female pelvic floor, but only segmented the levator ani muscle and the obturator muscles. The detailed anatomy of the superficial structures of the pelvic floor therefore remained unexplained [31]. Furthermore, in vivo scanning brings the risk of blurred images due to movements, for example from the pulsation of blood vessels or from bowel movements. Since some of the muscles in the pelvic floor are very thin, imaging resolution needs to be small enough to resolve their exact course and attachment sites. Histological preparation provides such high-resolution insight, but it is a destructive method and inherently (serially) two-dimensional. Translation into three-dimensional images is possible but rather difficult and time-consuming [32] and has only been done by very few authors so far [10,33–36].

X-ray microtomographic imaging allows high-resolution investigation of structures in a non-destructive way. Achievable voxel sizes are at least an order of magnitude smaller than those achieved by MRI [37] or ultrasound [38]. Biomedical samples are often immersed in contrast enhancing agents to stain different tissues and to make them distinguishable in tomographic scans [39]. A widely used contrast agent is iodine, mostly in form of iodine potassium iodide (Lugol). Metscher showed high-contrast images of soft tissues stained with iodine in different animal species [40]. Even delicate structures, such as the peri- and epimysium in rodent jaw muscles, can be made visible with Lugol staining prior to image acquisition [37]. Iodine staining has been used on a variety of species, such as various insects [40], vertebrate embryos [41], zebrafish [42] and mammals [37,43]. Usually, fixation is done for the least amount of time possible to enable further

processing, such as staining or embedding for histological sections. Little is known about the effect of prolonged fixation on the staining process. The use of long-preserved samples, such as those from anatomical collections, presents an opportunity to gather information without the need for new samples, which could be especially advantageous when studying human tissues.

The aim of this pilot study was I) to establish the method of µCT imaging of a Lugol-stained pelvic floor, to II) analyse the detailed female pelvic anatomy to contribute to the still controversial anatomical knowledge. III) We analysed the influence of the long fixation time of 4 years on the immersion protocol and subsequent tissue discrimination. To the best of our knowledge, this is the first time that the above-mentioned points are described in a paper.

This knowledge will enhance the anatomical understanding of the pelvis floor and support other scientists in using long-fixed specimens for Lugol immersion and imaging protocols. A detailed understanding of the pelvic floor anatomy is the basis for further clinical research.

## Materials & methods

### Material acquisition

For this study, the pelvis from a human female foetus was used. The specimen was provided by the Human Anatomy Department of the Orenburg State Medical University (Russia) from the Foetal Collection of the Department. The use of foetal human material was performed according to the Russian Federation Government Decree N.750 of 21 of July 2012 [44] and was regulated by the permission of the Ethical Committee of the Orenburg State Medical University. Each sample of foetal material was included into the foetal collection only after the mother provided written signed consent for its transfer for scientific purposes. This foetal collection includes foetuses without congenital defects which is confirmed by obligatory combined prenatal screening of the first trimester. Access of the data for research purpose was November 16th, 2017. The sample was completely anonymised, so that an identification was not possible for the researchers involved in the present study.

The gestational age was estimated at around 20–21 weeks, since the ossification centre of the pubic bone was already visible, which is the last to appear at around 5–6 intrauterine months [45] and by measuring the volume of the ossification centre in the ischial cartilage according to Baumgart et al. [46].

Additionally, the pelves of two adolescent mice were used to assess Lugol staining progression in relation to the duration of fixation. In accordance with the 3R principles, these pelves were sourced from another study, where the lower part of the body of the mice was not required [47]. This study was approved of by the Veterinary Office of the Canton of Bern and conducted under the Swiss license BE69/19. At 10 weeks of age, the C57BL/6J female mice were euthanized with an intraperitoneal injection of sodium pentobarbital (200 mg/kg BW).

### Fixation and Lugol immersion

The human pelvis had been immersed and stored in Formalin 8% (at 5° C) for approximately four years before being used in this study. It was in a well-preserved state and, macroscopically, no maceration was visible.

The mice were dissected after perfusion of the lower half of the body with 4% buffered Formalin (Biosystem Switzerland AG, CH-4132 Muttenz). The skin, the trunk at the height of the lower lumbar vertebrae and the lower legs were removed. Tissue fixation was performed by immersion in 4% buffered Formalin at room temperature for 24 hours and subsequent storage in the fridge at 5°C. The total fixation time for the mice pelves was two weeks.

Both human and mouse samples were subsequently rinsed with tap water for 30 min before immersing them in a Lugol solution. To avoid overstaining and to document the progression of the staining process, a 15% solution of Lugol Reagens diluted in *aqua dest.* was used (Artechemis AG, CH-4800 Zofingen). Lugol solution was replaced every 2–3 days, when its colour had changed to light yellow.

## µCT scanning and data analysis

The samples were imaged on a Bruker SkyScan 2214 high-resolution microtomography machine (Control software version 1.8, Bruker microCT, Kontich, Belgium). To document changes in tissue contrast over time, we acquired tomographic datasets of both samples approximately every 11 days over the course of approximately 5 months and 1 week for the foetal sample and 8 months for the mouse sample. The X-ray source was set to a tube voltage of 80.0 kV and a tube current of 120.0 µA and filtered by 1 mm of Aluminium for the foetal sample and to unfiltered 60.0 kV and 140.0 µA for the mouse sample.

The foetal sample was scanned with a geometrical setup resulting in an isometric voxel size of 20 µm, the mouse samples with a setup resulting in a voxel size of 15 µm. For each of the scans, projections with a size of 3072 x 1944 pixels at every 0.1° over 180° of sample rotation were recorded. At the final time point of the staining, high resolution scans of the samples were acquired with 11 µm for the foetal sample and 9 µm for the mouse sample, respectively.

All projection images were reconstructed into a stack of 8-bit PNG slices with NRecon (Version 2.1.0.1, Bruker microCT, Kontich Belgium). The grey values of the projection images were mapped into the reconstructions such that the contrast visible in the reconstructions was maximised into the 255 available grey values.

Data analysis was performed using a set of Jupyter [48] notebooks [49].

## Grey value in the samples over time

For each timepoint, the stack of reconstructions was separated into sample and background, based on a simple Otsu Threshold [50]. An average grey value for all the voxels in the sample volume was calculated and plotted (Fig 1A). The increasing average grey value follows a second-order polynomial regression, indicating an increase in contrast in the sample due to the more homogeneous and deeper Lugol staining.

## Muscle segmentation

Manual segmentation was performed on a binned dataset using 3D Slicer (version 5.2.2., https://www.slicer.org/, [51]). Relevant structures were manually segmented on approximately every third slice, complete regions of interest were interpolated with the "fill between the slices" function. Finally, the 3D model was edited by further smoothing the segmentation. The segmentation and the overlaid images, simultaneously visible in 3 planes, were used to get a 3D perception of the structures which allowed a detailed analysis of the muscles and their course.

## Results

### Tissue immersion and tissue differentiation

The mouse pelvis was smaller than the foetal one, with a dorsoventral diameter of approximately 1.6 cm and 2.9 cm, respectively, measured at the end of the staining process in both samples.

In the first scans, only the bony structures could be distinguished. In the foetal sample this was represented by the ossification centres in the different cartilage structures, which could – at this point – not be differentiated from the surrounding soft tissue (Fig 1B and 1E).

Over time, the iodine gradually diffused towards the centre of the specimen and the contrast in the soft tissues increased (Fig 1C and 1F). Tissue differentiation, as seen in the µCT scans, was optimal after around 5 months of Lugol immersion. Longer immersion of the mouse samples in Lugol did not result in increased tissue differentiation. Accordingly, the average grey value in the tomographic datasets was shown to increase with duration of Lugol staining and reached a plateau after about 100 days of immersion (Fig 1A). A final timepoint of tissue staining was recorded on day 158, after which there was no further increase in tissue differentiation (Fig 1A, 1D and 1G).

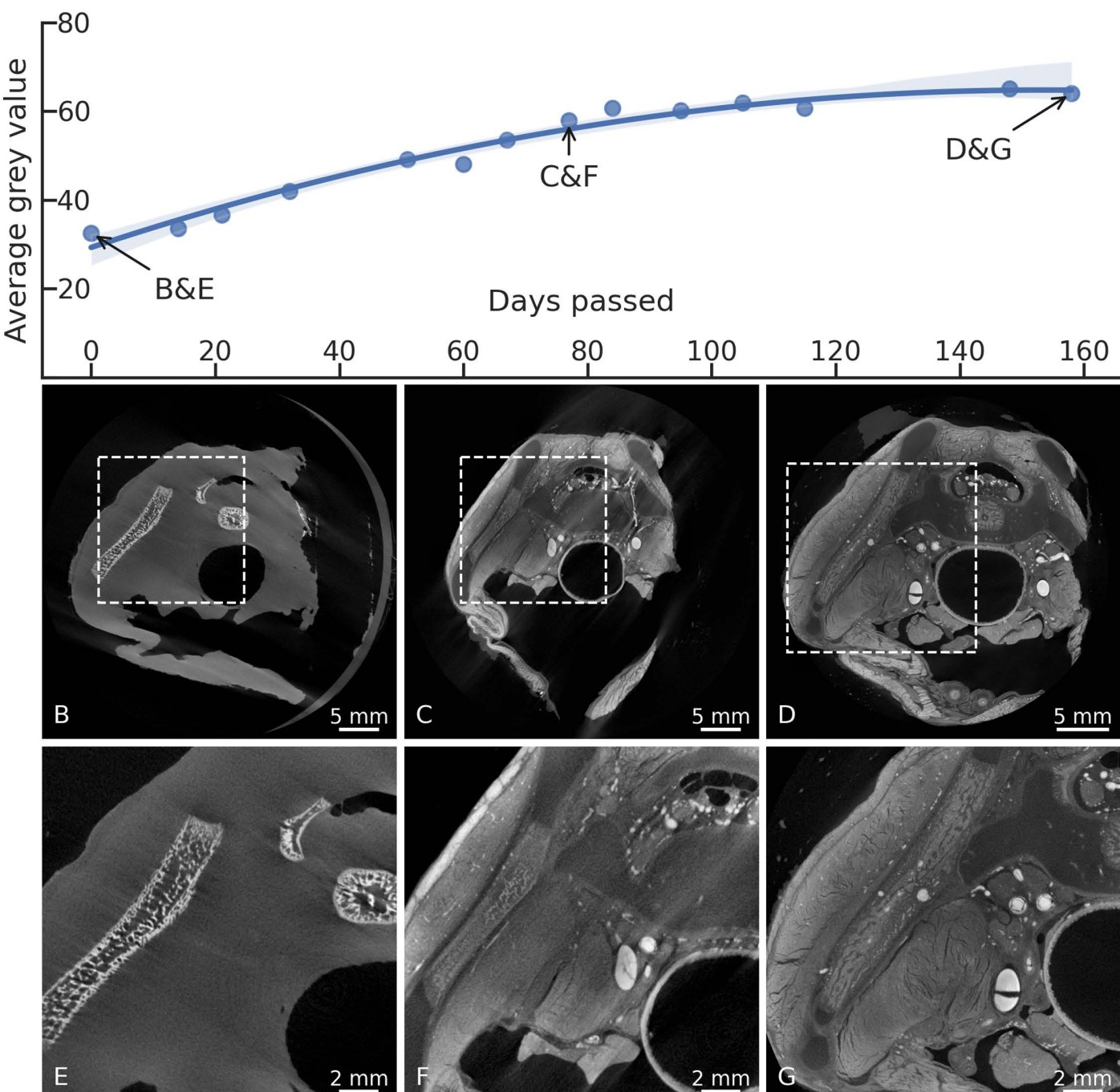

**Fig 1. Tissue differentiation and grey values over time.** The plot shows the average grey value over time for the foetal sample, calculated at different time points from all the voxels within the sample. Towards the end of the immersion period, a plateau can be observed. B), C), D) Overview images, showing the area where the detailed images E), F) and G) have been taken (dashed rectangles). E) Native scan of the human foetal pelvis, the ossification centres of the iliac bone (diagonally on the left) and two ossification centres of the sacrum (top right) can be seen. The soft tissue is not distinguishable. F) Scan after 77 days of immersion in Lugol, tissue differentiation is already increased, cartilage can be differentiated from bone, blood vessels are visible. G) The same area seen after 158 days of Lugol immersion. Besides the bony structures, now the cartilage, muscles, blood vessels and connective tissue can be distinguished. The full high-resolution tomographic dataset of the foetal sample at the last staining timepoint is available online for viewing (Supplementary material S1 File). The direct link to the region shown in panel G is https://webknossos.org/links/U8wuIdhmfzWMr7rm. Parameters of the tomographic scans are listed in Supplementary material S1 Table.

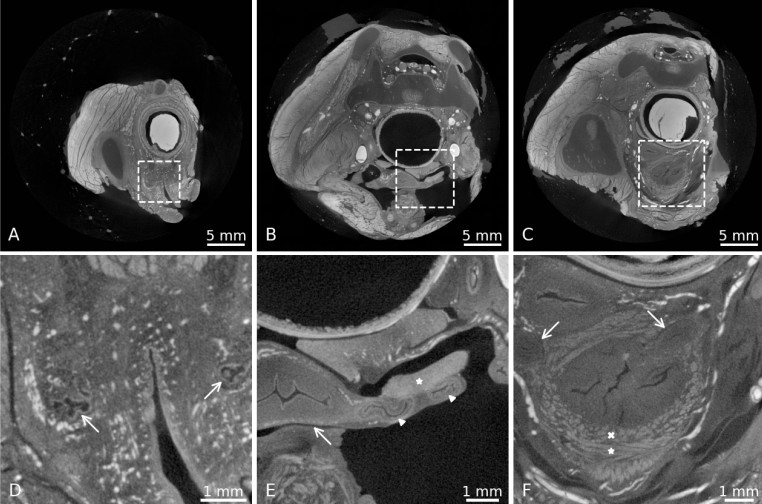

**Fig 2. Details of different anatomical structures in the human pelvis, seen after 158 days of Lugol immersion.** A), B), C) Overview images, showing the area where the detailed images D), E) and F) have been taken (dashed rectangles). D) Greater vestibular gland (arrow) on the left and right side of the vulvar vestibule. Direct link: https://webknossos.org/links/jJdvIlnaXk-opXhU E) Uterus (arrow) and several cuts through the left fallopian tube (arrowheads), parts of the ovary are also visible (*). Direct link: https://webknossos.org/links/FFDhAWFU2NlV6cLy F) Longitudinal (*) and transverse (x) view of the smooth muscle fibres within the bladder wall. The less contrasted mucosa can be distinguished from the muscularis layers. Arrows mark the ureter on both sides, passing through the bladder wall. Direct link: https://webknossos.org/links/c__8YJcw7K7e-_WV.

Finally, no influence of the long fixation time of the foetal pelvis (4 years) could be seen, compared with the shortly fixed mouse sample (2 weeks). The contrast between the different soft tissue components was enough to visually differentiate the various tissues and follow the course of the muscles in the three-dimensional datasets.

In general, iodine immersion allowed an easy differentiation of various tissues and organs within the pelvis. The different muscular, osseus, nerval and vascular structures, the female erectile tissue and its adjacent muscles of the pelvic outlet were recognizable. Even delicate structures, such as the greater vestibular glands (Fig 2A and 2D), the fallopian tube with its lumen (Fig 2B and 2E) or the smooth muscle fibres in the bladder wall (Fig 2C and 2F), could be seen in detail. The circular and longitudinal smooth muscle fibres of the rectum (Fig 3B, 3C, 3E and 3F) as well as the muscle fibre bundles in the bladder wall (Fig 2C and 2F) could clearly be seen in their course and distinguished from the mucosa and the connective tissue, but their appearance and brightness were very similar to striated muscles, such as the BS (Fig 3A), the EAS (Fig 3C and 3D) or the internal obturator muscle. A morphologic differentiation between smooth muscle and striated muscle tissue was not possible but muscle identification could be done by knowing the course of the respective muscle. Within the smooth muscle tissue, distinct areas could be defined, based on the intensity of the contrast and the visibility of muscle fibre bundles. 1. dense areas with clearly visible fibre course (e.g., longitudinal smooth muscle of the rectum, bladder wall, Fig 3C and 2F), 2. dense areas with no visible fibre course, likely three-dimensionally arranged fibres (e.g., the region of the DTP, Fig 3B and 3C)) and 3. sparse areas, that were lower in contrast compared to the other two, but still corresponded to areas with presence of smooth muscle tissue (e.g., longitudinal smooth muscle of the rectum, interface between levator ani and pelvic organs, Fig 3C). Connective tissue appeared darker (less contrasted) in the µCT images and could therefore be distinguished from the muscular structures.

After the immersion in Lugol, the mouse pelves showed some signs of maceration, with the bones becoming soft and elastic. The human pelvis, that had been fixed for a much longer time, showed no signs of degradation.

### Anatomy of the pelvic outlet region

**Bulbospongiosus and ischiocavernosus muscle.** The BS surrounded the bulbus vestibuli inferiorly (Fig 3A) and could be tracked in its course towards the STP and the EAS. On both sides, there were identifiable fibres running from the

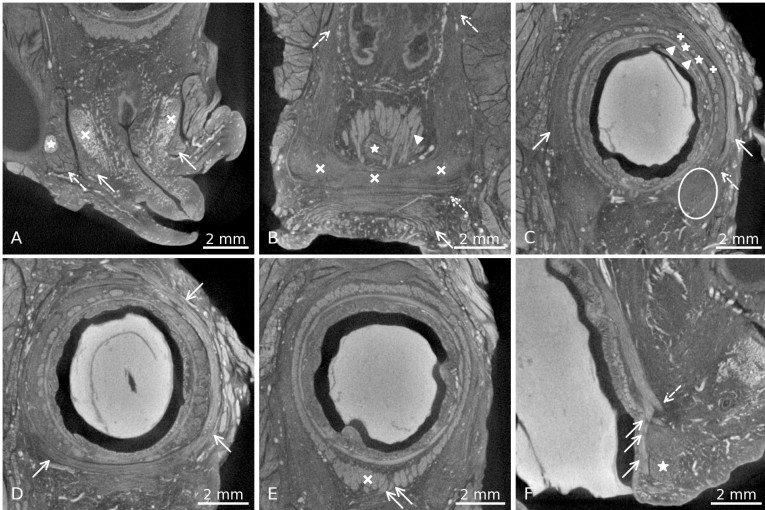

**Fig 3. Overview images of pelvic floor muscles.** A) Transverse section, arrows indicate the left and right BS, covering the vestibular bulb on both sides (x). Dashed arrow shows the ischiocavernosus (IC) muscle on the right side, covering the crus of the clitoris (*). Direct link: https://webknossos.org/links/WXhDBfVxvyFCfoWV B) Coronal section through the anterior wall of the rectum. Arrow: EAS fibres, dotted arrow: STP, X's: cloudy looking area of the DTP, dashed arrows: LA muscle, caudally extending into the area of the DTP, arrowhead: LM fibres, (*): transverse smooth muscle of the rectum. Direct link: https://webknossos.org/links/e9Rjlm0bgCA8T8C0 C) Transverse section through the rectum. The arrows indicate the EAS fibres in their course around the smooth muscles of the rectum. At the dashed arrow, the merging with the DTP can be observed, the DTP area on the left side is encircled. Within the LM fibres, dense (*) and sparse (+) areas can be distinguished. The transverse fibres of the rectum are marked with arrowheads. Direct link: https://webknossos.org/links/wX2zNqTjo0FtNAFl D) Superficial transverse section through the rectum, arrows show the circular course of the EAS fibres around the rectum smooth muscles. Direct link: https://webknossos.org/links/R1nf3dITqxCbHcHL E) Transverse section through the rectum, showing the circular and longitudinal smooth muscle layers. White x shows the area of thickening of LM fibres in the ventral wall of the rectum/ anal canal. Arrows indicate fibres coursing anteriorly towards the perineal centre. Direct link: https://webknossos.org/links/3oPQxNHvZuHAoH7C F) Sagittal section through the anterior wall of the rectum. Arrows show the LM fibres coursing posteriorly to the EAS (*) towards the anal skin, dashed arrow shows the anterior bundle, running more anteriorly and dispersing towards the perineal centre/ DTP area (arrows in E). Direct link: https://webknossos.org/links/GTY_1YrY7_YsBF0k.

BS into the EAS, and some fibres from the BS ran into the STP, forming one unit (Fig 4). It could not be unambiguously defined if the BS fibres crossed within the PB form a sling with the opposite BS, or just ran into it.

The ischiocavernosus muscle (IC) surrounded the crura of the clitoris (Fig 3A) and ran laterally and inferiorly along the inferior pubic branch towards the ischial tuberosity, without connection to the centre of the pelvic floor.

**Superficial transverse perineal muscle (STP).** The muscle appears as a delicate bundle of fibres running in a transverse direction, caudal to the area of the DTP and posteriorly to the BS muscle (Fig 3B). There was no visible attachment to the cartilaginous frame of the pelvis laterally. Instead, antero-posteriorly running blood vessels separated the lateral extensions of the STP from the inferior pubic ramus. On both sides, as described above, the muscle merged into the BS with some of its anterior fibres (Fig 4). Some posterior extensions towards the EAS could also be seen. There was a smooth transition between the STP and the most cranial circular fibres of the EAS, it was not possible to define a clear border between the two muscles (Fig 5).

**Deep transverse perineal muscle (DTP).** The DTP could not be distinguished as a distinct skeletal muscle but appeared as a brighter, cloudy area with some short, transversely running fibre bundles within it (Fig 3B and 3C). It was located cranially to the STP and postero-medially to the cranial parts of the BS and had no fibres running into either of them. Fibres from the LM (Fig 3E), the EAS (Fig 3C) and the levator ani could be seen merging with this cloudy-looking tissue (the LA extending from cranially into it, Fig 3B). The contrast of the tissue was increased in comparison to connective tissue areas in the scan, corroborating the assumption that smooth muscles were intermingled with the

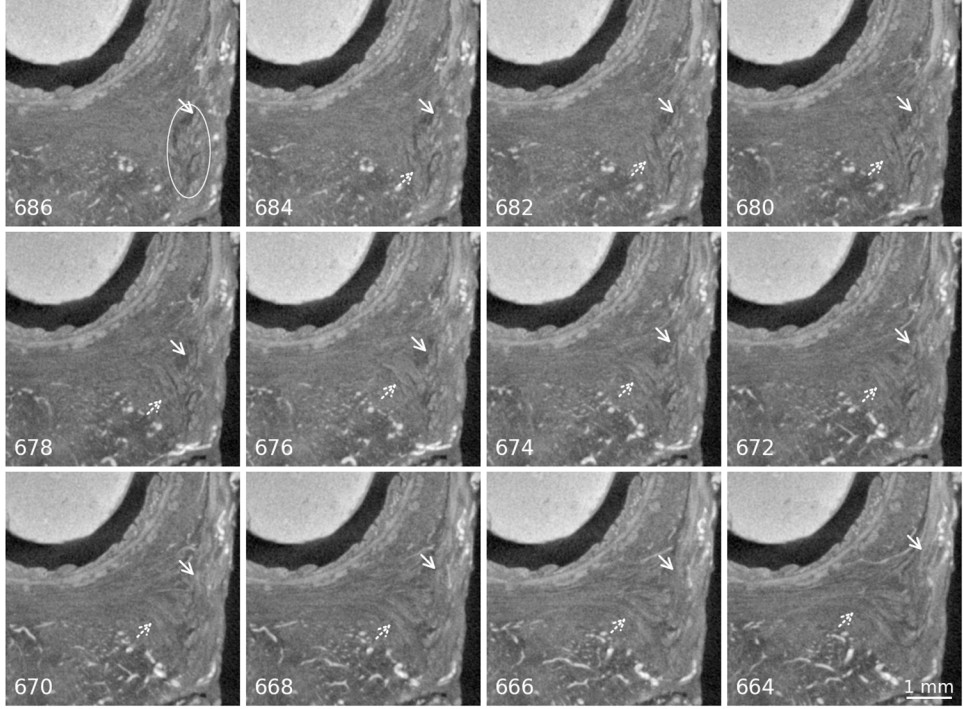

**Fig 4. Merging of the BS with the EAS and the STP.** Panels are labelled with the slice number of the image in the original tomographic stack, distance between the slices is 22 µm. Scale bar valid for all images. In the first picture the BS muscle fibres are encircled in white, arrows indicate the BS fibres running into the EAS. Dotted arrows indicate the BS fibres running into the STP. Direct link to region in slice number 676: https://webknossos.org/links/0JfbMABUk4AWsH8I.

connective tissue (as described in the previous section). There was no visible connection towards the cartilaginous frame of the pelvis.

**External anal sphincter (EAS).** The EAS was observed as a muscle with a horse-shoe shape (open anteriorly) in its cranial parts, surrounding the IAS and merging with the cloudy plate in the area of the DTP (Fig 3C). Some fibres of the EAS also ran into the BS and the STP (Figs 4 and 5). Cranially, the EAS was continuous with the LA without a clearly defined border between the two. Caudally to the DTP and the STP, the fibres of the subcutaneous layer of the EAS became circular around the anus and were visible also in the very superficial areas towards the anal skin (Fig 3D).

**Rectum smooth muscles.** Both smooth muscle layers of the rectum had insertions into the area of the perineal centre, longitudinal fibres (longitudinal muscle, LM) more cranially, transverse fibres (TM) more caudally. Most fibre bundles scattered cranially to the DTP area and within it, but a small number of the longitudinal fibres could be seen extending caudally towards the skin and through the subcutaneous part of the EAS (Fig 3F). Some other anterior longitudinal fibres could be observed to course anteriorly to the EAS (Fig 3E and 3F).

Within the LM, two distinct characteristics of the muscle could be seen. The first one presented as clearly defined muscle bundles, whereas the second one was less dense, seemingly more intermingled with connective tissue (Fig 3C). Cranially, those sparse areas were arranged circularly around the denser areas of LM and connected to the LA. Caudally, the dense and sparse tissues intermingled without merging, making the dense areas more clearly visible as muscle bundles.

**Levator ani muscle.** The LA could be recognised at its origin on the posterior aspect of the pubic symphysis and could be tracked in its course around the rectum. Interestingly, there seemed to be a connection between the LA to the LM of

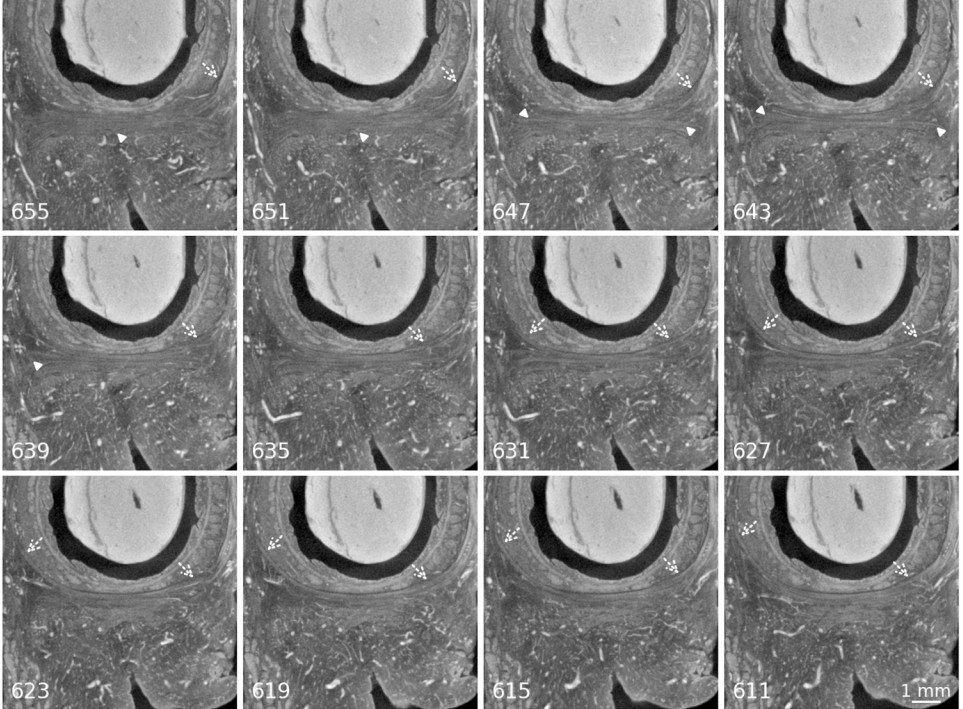

**Fig 5. Interface between the STP and the EAS.** Panels are labelled with the slice number of the image in the original tomographic stack, distance between the slices is 44 µm. Scale bar valid for all images. Arrowheads show the STP and its extensions towards lateral. Dotted arrows point at the EAS and the merging of its fibres with the STP. In the first slices this is visible only on the left side, towards caudal also on the right side. In the third row, the STP has disappeared from the image plane and the circular fibres of the EAS can be seen. Direct link to slice number 635: https://webknossos.org/links/sIhFR0JEpMcvLgn9.

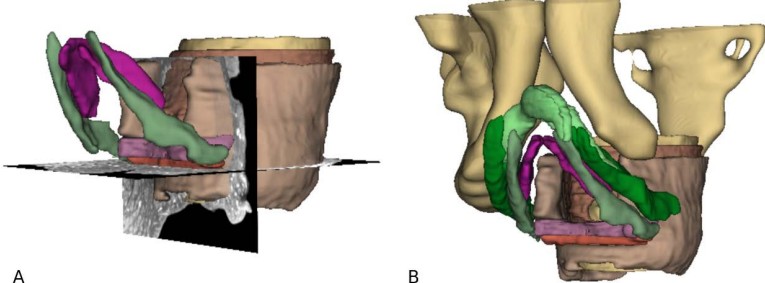

A                B

**Fig 6. 3D-reconstruction of the pelvic outlet muscles based on µCT images, view from anterior left.** A) Shows the rectum with its circular (yellow) and longitudinal (brown) smooth muscle layer, as well as the EAS surrounding it (light brown). The bulbus vestibuli is shown in purple, BS muscle in green. In pink, the DTP is visualised, caudal to it in orange the STP. Additionally, 2 planes from the original stack can be seen. B) Additionally to A), the crura of the clitoris with the ischiocavernosus muscle can be seen in light and dark green, respectively. The cartilaginous frame of the pelvis (inferior pubic rami, right iliac and ischial bone and sacrum) is shown in yellow.

the rectum, not in the form of clearly defined muscle bundles, but rather as a coalescing connective tissue/smooth muscle structure. The same could be observed between the LA and the smooth muscle tissue in the region of the DTP/central region of the perineum (Fig 3B).

The origin of the LA from the fascia of the internal obturator muscle (arcus tendineus musculi levatoris ani) was clearly visible in the dataset, as well as the insertion posterior to the anal canal in the anococcygeal raphe (also called levator raphe or anococcygeal ligament).

## Segmentation of the pelvic outlet muscles

Manual segmentation of the muscles in the pelvic floor and subsequent 3-D reconstruction helped to visualise the topographical relations between the muscles and corroborated the findings from the analysis of the 2-dimensional µCT images (Fig 6). Unfortunately, it was technically not possible to overlap segmented areas, meaning that the interconnection of the different muscles could not be depicted precisely in the 3-D model. Nevertheless, following structures simultaneously in 3 planes helped to grasp the pelvic outlet region as a whole and could serve as an additional source of information.

To summarise, there were fibres from the EAS, the BS and the STP running through the centre of the perineum, while the smooth muscle from the rectum ran from superiorly towards it, probably providing fibres for the area. There was no evidence for a skeletal muscle plate that could correspond to the DTP, instead there was cloudy-looking tissue, most likely smooth muscle fibres dispersed in a 3-D arrangement, but not in the form of fibre bundles that were important enough to be resolved in the µCT images. The interface between LA and LM, thus between skeletal and smooth muscle, could be brought out, as well as the different features of the LM (dense and sparse areas). In the 3D reconstruction, the topography of the region could be illustrated.

## Discussion

The pelvic floor is crucial for the function and health of the lesser pelvis and its organs by providing stability and continence. Unfortunately, there is still a controversial understanding of the exact anatomy of the pelvic floor muscles in females.

Women are much more affected by pathologies regarding the pelvic floor. Prevalence for at least one pelvic floor disorder (urinary incontinence, faecal incontinence, or pelvic organ prolapse) [1] is high, and even nulliparous women may present a substantial risk to develop POP [5] The severity and symptoms of POP can be improved by training of the pelvic floor muscles [6]. Thus, the exact description of the healthy, in situ anatomy could be the basis for a further understanding of the above-mentioned conditions. Knowing about the healthy anatomical connections in the pelvic floor muscles could help extend the knowledge about the consequences of a lack of integrity in these structures.

In this study, the well-established method of contrast enhancement with Lugol staining [39] has been used to show the detailed structures of the foetal pelvic floor in situ. Even though the sample had previously been immersed in fixative for several years, no degradation in µCT-image quality was observed (Figs 2 and 3). This could be explained with the cross-linking of proteins that remains stable also after years of immersion [52]. Furthermore, the progression of iodine staining seemed independent of the former fixation time, as it was comparable to a mouse pelvis fixed only for two weeks. In the shortly fixed mouse sample, though, there was a slight degradation of the tissue towards the end of the staining process. Probably, to enable the immersion in a watery solution for such a long time, the fixation of the tissue should have been longer to prevent this from happening. Another possibility to avoid maceration could be the perfusion of the samples with Lugol to gain contrast in a shorter time-period, a method that has been established by Witmer et al. [53]. A female foetus of approximately 20 weeks of gestation was chosen to demonstrate the anatomy, representing tissue not affected by training, age, or birth.

As explained in the introduction, there are still some open questions regarding the anatomy of the pelvic floor in females. Firstly, regarding the existence of the DTP, this study revealed no evidence for a skeletal muscle in the previously described area of the muscle. The images from the present study showed the area of the expected DTP to be consisting of tissue with a contrast between connective tissue and muscle tissue, leading to the assumption that smooth muscle fibres were dispersed within the connective tissue. This could not be defined unambiguously in the µCT images, but it

corroborated several previous studies [10,24,54]. In their review, Muro and Akita presented a new understanding of the pelvic outlet with a dynamic interplay between smooth muscle and skeletal muscle, describing the DTP as a smooth muscle structure [24]. This view was also supported by histological studies that examined the tissue composition of the area between rectum and vagina and found smooth muscle and connective tissue to be a major component of it [35,55,56].

The lack of visibility of those smooth muscle fibres could be explained by the fact that they did not present as well-defined fibre bundles, as in the rectum or the bladder wall, but were arranged in a 3D-manner between the connective tissue in the central region of the pelvic outlet. This finding correlates with the description of Muro et al. of dense and sparse areas within smooth muscle tissue, as seen in MRI and on histological sections [57]. Nevertheless, the connective tissue in the region of the DTP was more contrasted than in other areas, suggesting a strong likelihood that smooth muscle tissue was also present.

The origin of this smooth muscle tissue has been the subject of several studies. Fritsch et al. and Sebe at al. found that the longitudinal smooth muscle of the rectum (LM) extended to surrounding structures at 8–9 weeks of the embryonic development, suggesting that they might form parts of the smooth muscle in the perineum [58,59]. Consistently, in this study, LM fibres could be seen to extend to the area of the perineal centre (Fig 3E and 3F), which has already been reported previously [57,60,61]. They described its diffuse extensions into the perineum anteriorly to the EAS (the region of the perineal body), some of the authors calling those fibres the "anterior bundle", others the "conjoint longitudinal coats". This could be an explanation for the origin of the smooth muscles that form the so-called DTP [24]. Taken together, we would suggest that what is classically referred to as the DTP in literature should be reframed to acknowledge its 3-D arrangement and its composition of smooth muscle and connective tissue. Possibly, the term perineal membrane could be used to describe the region of the DTP more accurately.

Secondly, regarding the question of a crossing of fibres in the PB, the following observations could be made in this study: The course of the BS, EAS and STP, as well as the longitudinal and transverse fibres from the rectum wall towards the area of the perineal body, could be observed. In the present study, the BS had fibres extending posteriorly into the EAS of the same side (Fig 4). This is in accordance with Baramee et al. and Muro et al. and was explained by the latter with the fact that the female BS is divided on both sides of the vaginal opening, as is the bulb of the vestibule [19,24]. Additionally, the BS had some fibres that ran medially towards the STP, melting with it. The STP presented as a delicate, striated fibre bundle with crossing fibres through the area of the perineal body without interruption, which has also been confirmed by other authors [19,24]. In accordance with prior studies, no attachment of the STP to the pelvic ring was visible [10,11]. This stands in contrast to the findings of Baramee et al., who described the attachment of the STP to the ischial tuberosity [19]. An explanation for this discrepancy might be the stage of development of the foetus used in the present study. The extensions towards the lateral pelvic wall might develop later, possibly even after birth with the onset of walking and thus gravity working on the muscles of the pelvic outlet.

The EAS could be seen as a continuous structure without clear borders to the LA. This finding could be due to the overlapping of those two muscles, as it was described by Tsukada et al. and Zang et al. [62,63]. Other authors also found a continuity between the puborectalis (as part of the LA) and the EAS [64,65]. Arakawa et al. even described different morphologies of the merging of LA fibres with the rectal smooth muscles [66]. According to Baramee et al., the exact anatomical relationship between the EAS and the STP are still unclear [19]. This study revealed the position of the STP anteriorly to the EAS, lying between the cranial, horse-shoe shaped portion of the EAS and the caudal, circularly arranged portion of it. The cranial parts of the EAS merged with the tissue in the area of the DTP (Fig 3C). In the circular part there was no clear border towards the STP, some fibres of the STP seemed to run posteriorly and merge with the EAS (Fig 5). Zifan et al. described crossing fibres from the EAS into the BS of the opposite side [20]. Since it was not possible to manually follow single fibre bundles over such a long course, this arrangement could neither be confirmed nor denied in the present study. But it could well be that the above-mentioned merging fibres between EAS and STP continued their course anteriorly and into the BS, which was corroborated by the observation of fibres connecting the BS and the STP. As it has also been mentioned by Guo et al., the subcutaneous (circular) parts of the EAS had a very superficial position, almost reaching the anal skin [67].

Thirdly, this study could not reveal a clear entity that could be defined as the PB, but it clearly showed that the area of the pelvic outlet is a network of both smooth and skeletal muscles with crossing fibres, intermingling and merging into each other, providing a dynamic support for this region. It cannot be seen as an aggregation of singular muscles, attaching at a specific point. This interconnectedness was also described by Kato et al., who found smooth muscle tissue as an interface between the levator ani and the pelvic viscera [68].

Clearly, there are some limitations to this study. Firstly, only one foetal sample was used for the description of the anatomy, which might vary between individuals. Further studies including more samples, also from other stages of development, could increase the accuracy of our findings. Various other factors, such as giving birth, hormonal status and aging are known to influence the structures of the pelvic floor, so in the future, more studies are needed to document these changes and compare them to the healthy anatomy. Nevertheless, as described above, other studies found similar anatomical relationships. Due to the more precise description of the pelvic muscles and their non-destructive assessment in high-resolution three-dimensional data (which is available to other researchers) our results help to advance the understanding of the interplay of the pelvic muscles in detail. The resolution of µCT images is lower than in histological sections, so very small muscle fibres could be missed in the analysis and lead to false conclusions regarding the course of the observed muscles. However, the resolution is still around an order of magnitude higher than in MRI or ultrasound [11,30,38] and single muscle fibre bundles can be resolved. Even the combination method of Muro et al., combining serial sectioning with 3D-reconstruction, had a distance of 1 mm between the histological sections [36]. We deliberately did not perform a histological assessment of our samples, which could have confirmed our assumptions of the tissue composition within the perineal membrane. Firstly, other researchers have already performed similar studies with the same results, and secondly, we did not want to resort to a destructive analysis method. Furthermore, the contrast of connective tissue is different from (smooth) muscle, so the grey value in the µCT images gives a good information about the tissue composition in a certain area. Due to the potentially inhomogeneous diffusion of contrasting agents into the tissues, another possible limitation lies in the sample size, which could be challenging in samples from adult persons.

Strengths of this study are in its non-destructive method and the high resolution that could be obtained with µCT scans, allowing to distinguish even delicate muscle fibre bundles. The 3-D model generated from the images show the topographical relationships of the different muscles. This could be useful to gain a more accurate understanding of the pelvic outlet region. Additionally, by using a foetal sample, factors that could have changed the tissues of the pelvic floor over time were excluded, showing its unaltered anatomy.

## Conclusion

The detailed anatomy of the female pelvic floor of a human foetus was demonstrated using contrast-enhanced µCT imaging. The results from imaging and segmentation of the muscles corroborated previous findings in the literature, showing that this method could be valuable for the description of muscular anatomy. µCT imaging is an excellent tool to non-destructively visualize anatomical relationships. This study confirmed the outcome of previous research that the female pelvic floor should rather not be seen as singular muscle entities, but as a 3D arrangement of continuous, intertwined skeletal and smooth muscles, with the so-called DTP likely forming a smooth-muscle structure.

## Supporting information

**S1 File. Full high-resolution tomographic dataset of the foetal sample.** Due to the file-size, the acquired dataset at the last staining timepoint is only available online for viewing at https://webknossos.org/datasets/Institute_of_Anatomy/Foetus02_Lugol_15pct_152d_rec/view This is the dataset from which we extracted the images shown in Figs 1D, 1G and 2–5. Direct links to the regions shown in the panels are given in the figure captions.
(DOCX)

**S1 Table. Scan parameters extracted from logfiles.** Relevant data of the log files (https://github.com/habi/FemalePelvicFloor/tree/main/logfiles) of the scans shown in Fig 1.
(XLSX)

**S2 File. Inclusivity in global research questionnaire.**
(DOCX)

## Acknowledgments

We thank Eveline Yao for her excellent work in the laboratory. Both the input from Dr. Alexander Ernst and Oleksiy-Zakhar Khoma was vital for performing the segmentation of the structures of interest. Prof. Dr. med. Valentin Djonov provided guidance for this work. Dr. med. Nikola Tomov provided inspiration for this project.

## Author contributions

**Conceptualization:** Dea Aaldijk.

**Data curation:** David Haberthür.

**Formal analysis:** Dea Aaldijk, Sebastian Halm.

**Investigation:** Dea Aaldijk, Sebastian Halm.

**Methodology:** Dea Aaldijk, David Haberthür.

**Project administration:** Dea Aaldijk.

**Resources:** Diana N. Liashchenko.

**Software:** David Haberthür.

**Supervision:** Dea Aaldijk.

**Validation:** Dea Aaldijk, Sebastian Halm, Diana N. Liashchenko.

**Visualization:** Sebastian Halm, David Haberthür.

**Writing – original draft:** Dea Aaldijk.

**Writing – review & editing:** Dea Aaldijk, Sebastian Halm, Diana N. Liashchenko, David Haberthür.

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
