## [Decision Letter · Decision Letter 0]

12 Feb 2025

PONE-D-24-49109Contrast-Enhanced Micro-CT Imaging of a Foetal Female Pelvic Floor Reveals Anatomical DetailsPLOS ONE

Dear Dr. Aaldijk,

Thank you for submitting your manuscript to PLOS ONE. After careful consideration, we feel that it has merit but does not fully meet PLOS ONE’s publication criteria as it currently stands. Therefore, we invite you to submit a revised version of the manuscript that addresses the points raised during the review process.

We look forward to receiving your revised manuscript.

Kind regards,

Stanisław Jacek Wroński, M.D., Ph.D, FEBU

Academic Editor

PLOS ONE

Journal Requirements:

Additional Editor Comments:

Dear Authors,

having considered the submitted paper entitled "Contrast-Enhanced Micro-CT Imaging of a Foetal Female Pelvic Floor Reveals Anatomical Details" and having considered the reviews, I suggest a major revision of this submission.

I would ask you to consider the following comments:

1. Why was only one fetal sample used? How might anatomical variation affect the results?

2. How might the years of fixation impact tissue properties and the accuracy of µCT imaging?

3. How do you differentiate between smooth muscle and connective tissue in µCT scans? Can histological analysis support your findings?

4. Could µCT imaging miss smaller muscle fibers or fine details? How might this impact the results?

5. How might variations in age, hormonal status, or medical history affect pelvic floor anatomy in other individuals? you only have a single sample

6. How can your anatomical findings directly impact the understanding and treatment of pelvic floor disorders like POP?

7. What are the limitations of µCT for visualizing soft tissues, and how do these affect the findings?

8. How do your results compare with those from MRI or ultrasound in terms of resolving pelvic floor anatomy?

9. Have similar studies using µCT or other imaging methods confirmed your findings in different samples? Note: all the answers to the former questions MUST be included in the Study Limitations.

10. Do you think the term "DTP" needs clearer definition, given the ongoing debate about its structure?

11. The authors should mention the term 'crossing fibers' in the perineal body, similar to the findings established in the paper by Zifan et al in the perineal region. This has already been well-documented using DTI and micro-CT imaging.

12. Given that only one sample was used, the authors need to tone down their conclusions to reflect speculation rather than definitive findings.

Hope you will find above remarks helpful in adapting your work to the requirements and level of PLOS ONE

with compliments

Stanisław Wroński

Academic Editor

Reviewers' comments:

Reviewer's Responses to Questions

**Comments to the Author**

1. Is the manuscript technically sound, and do the data support the conclusions?

Reviewer #1: Partly

2. Has the statistical analysis been performed appropriately and rigorously? 

Reviewer #1: N/A

3. Have the authors made all data underlying the findings in their manuscript fully available?

Reviewer #1: Yes

4. Is the manuscript presented in an intelligible fashion and written in standard English?

Reviewer #1: No

5. Review Comments to the Author

Reviewer #1: Why was only one fetal sample used? How might anatomical variation affect the results?

How might the years of fixation impact tissue properties and the accuracy of µCT imaging?

How do you differentiate between smooth muscle and connective tissue in µCT scans? Can histological analysis support your findings?

Could µCT imaging miss smaller muscle fibers or fine details? How might this impact the results?

How might variations in age, hormonal status, or medical history affect pelvic floor anatomy in other individuals? you only have a single sample

How can your anatomical findings directly impact the understanding and treatment of pelvic floor disorders like POP?

What are the limitations of µCT for visualizing soft tissues, and how do these affect the findings?

How do your results compare with those from MRI or ultrasound in terms of resolving pelvic floor anatomy?

Have similar studies using µCT or other imaging methods confirmed your findings in different samples?

Note: all the answers to the former questions MUST be included in the Study Limitations.

Do you think the term "DTP" needs clearer definition, given the ongoing debate about its structure?

The authors should mention the term 'crossing fibers' in the perineal body, similar to the findings established in the paper by Zifan et al in the perineal region. This has already been well-documented using DTI and micro-CT imaging.

Given that only one sample was used, the authors need to tone down their conclusions to reflect speculation rather than definitive findings.

6. PLOS authors have the option to publish the peer review history of their article (what does this mean? ). If published, this will include your full peer review and any attached files.

**Do you want your identity to be public for this peer review?** For information about this choice, including consent withdrawal, please see our Privacy Policy .

Reviewer #1: No

---

## [Author Response · Author response to Decision Letter 1]

8 Apr 2025

Dear Academic Editor, dear Reviewer

Thank you for your valuable feedback and your important questions aiming at making our manuscript more precise and adequate. In the following text, each of your comments will be answered separately.

Journal Requirements:

The text has been formatted according to the guidelines as accurately as possible.

As requested, a completed version of the questionnaire is uploaded together with the updated manuscript.

3. We note that your Data Availability Statement is currently as follows: [All relevant data are within the manuscript and its Supporting Information files.

Yes, all relevant data are within the manuscript and its Supporting Information files. The Jupyter notebooks to perform the analysis are available online and linked in the manuscript. The dataset from which the views in the figures are generated is available online. Due to limitations of data size, the datasets to calculate the average gray value shown in Fig. 1A are not available online, but on request. The calculated average gray values are listed in the supplementary materials.

Additional Editor Comments:

Since those additional comments consist of the same questions as raised by the reviewer, they are addressed below.

Reviewers' comments:

Reviewer's Responses to Questions

Comments to the Author

1. Is the manuscript technically sound, and do the data support the conclusions?

Reviewer #1: Partly

2. Has the statistical analysis been performed appropriately and rigorously?

Reviewer #1: N/A

3. Have the authors made all data underlying the findings in their manuscript fully available?

Reviewer #1: Yes

4. Is the manuscript presented in an intelligible fashion and written in standard English?

Reviewer #1: No

Dear reviewer, please indicate us what kind of corrections need to be done, or which passage should be more comprehensible, so that we can address this issue more precisely. Nevertheless, we read through the manuscript again and tried to improve the grammar.________________________________________

5. Review Comments to the Author

Reviewer #1:

1. Why was only one fetal sample used? How might anatomical variation affect the results?

Thank you for this comment. Of course, anatomical variation might affect the results, and further scanning of fetuses (also of various stages of development) might help to verify our findings or define possible variations. A sentence addressing this was added at lines 499-503. The reason to use only one sample was to explore the feasibility of using long-stored samples for image analysis and to assess image quality as part of a pilot study. Furthermore, procurement of another sample for investigation is currently impossible.

2. How might the years of fixation impact tissue properties and the accuracy of µCT imaging?

The accuracy of the imaging itself is not influenced by the duration of fixation. The possible impairment in tissue properties was one of our research questions: therefore, we compared the image quality of the foetal sample with a mouse sample fixed for only two weeks. Even though the muscular anatomy in the mouse is different, changes in tissue quality, and therefore possibly less visibility of structures, could have been observed but were not present. We explain this with the fact that the crosslinking of proteins induced by the fixation with Formalin remains stable over the years and does not further impact tissue quality. However, a slight degradation of the mouse tissue that had been fixed only for two weeks was observed. This may be attributed to immersion in a watery solution with Lugol for a long time combined with the insufficient fixation time to ensure long-term stability. In lines 419-427, a description and explanation were added.

3. How do you differentiate between smooth muscle and connective tissue in µCT scans? Can histological analysis support your findings?

The differentiation is based on the grey values and the presence or absence of fibre-like structures. In the µCT scans, connective tissue appears darker and less contrasted than muscle. Histological analysis, as it has already been done by other authors, supports our observations/classification of the different tissue types. An additional sentence explaining this has been added in the results section, lines 254-255. We additionally mentioned this in the limitations of our study at lines 516-518.

4. Could µCT imaging miss smaller muscle fibers or fine details? How might this impact the results?

Of course, the limitations lie in the resolution of the µCT images, which could potentially lead to a false interpretation of the course of some muscle fibres. Nevertheless, with a voxel size of 10 µm, the resolution of our images is much better than in previous MRI and other three-dimensional imaging studies. While histological sectioning offers higher resolution, it comes with the drawback of being (serially) two-dimensional and destructive to the sample. Our non-destructive approach allows us to track single fibre bundles throughout the entire image stack. Additional explanations regarding the resolution of the µCT imaging have been added at lines 507-512.

5. How might variations in age, hormonal status, or medical history affect pelvic floor anatomy in other individuals? you only have a single sample

Thank you for this question. Of course, various factors influence pelvic floor anatomy. Our aim was to show the unaltered anatomy of a pelvic floor to have a reference to compare with. Future studies should investigate the effects of aging, hormonal status, medical history, childbirth as well as other factors on pelvic floor integrity and anatomy. This was answered together with question 1 at lines 499-503.

6. How can your anatomical findings directly impact the understanding and treatment of pelvic floor disorders like POP?

Thank you for this question. A direct impact on the treatment options for pelvic floor disorders is, in fact, difficult to foresee or describe at this state of research. We have revised the relevant paragraph in the discussion on lines 412-415, focusing on the importance of having a healthy model as a foundation for further research on pathological conditions.

7. What are the limitations of µCT for visualizing soft tissues, and how do these affect the findings?

Limitations for the visualization of soft tissues in µCT are given by the fact that discrimination is not possible in unstained tissues. Many studies have already shown the effect of contrasting agents on the differentiation of various soft tissues, as cited in the introduction (lines 110-119). The main limitation stems from the potentially inhomogeneous diffusion of the contrasting agent into the tissue (for large samples). A sentence has been added to the study limitations at lines 518-520.

8. How do your results compare with those from MRI or ultrasound in terms of resolving pelvic floor anatomy?

This question has been addressed in the introduction section (lines 110-112) as well as in a new section in the discussion, lines 507-512. Since this closely relates to question 4 we explain this together in our discussion section.

9. Have similar studies using µCT or other imaging methods confirmed your findings in different samples?

Similar studies using µCT and other imaging techniques have provided valuable insights into soft tissue differentiation and anatomical structures. While our study focuses on a specific sample, previous research has demonstrated the feasibility of µCT for high-resolution imaging of soft tissues, particularly with contrast enhancement. We cite several of these studies in our manuscript both in the introduction and discussion. To the best of our knowledge, our study is the first one to show the muscular anatomy of the female pelvic floor in contrast-enhanced µCT images.

Note: all the answers to the former questions MUST be included in the Study Limitations.

As mentioned in the single questions, all answers are provided in the discussion section in the study limitations, lines 498-520.

Do you think the term "DTP" needs clearer definition, given the ongoing debate about its structure?

Yes, thank you for this suggestion, we added this in lines 456-459.

The authors should mention the term 'crossing fibers' in the perineal body, similar to the findings established in the paper by Zifan et al in the perineal region. This has already been well-documented using DTI and micro-CT imaging.

The term has been added to the discussion at lines 468 and lines 484-489.

Given that only one sample was used, the authors need to tone down their conclusions to reflect speculation rather than definitive findings.

Thank you for this important advice. The respective sentences in the conclusion were adjusted accordingly, lines 528-535.

Line numbers given in the answers above correspond to the document with track changes where the adjustments are highlighted.

---

## [Decision Letter · Decision Letter 1]

27 Apr 2025

Contrast-Enhanced Micro-CT Imaging of a Foetal Female Pelvic Floor Reveals Anatomical Details

PONE-D-24-49109R1

Dear Dr. Dea Aaldijk

We’re pleased to inform you that your manuscript has been judged scientifically suitable for publication and will be formally accepted for publication once it meets all outstanding technical requirements.

Kind regards,

Stanisław Jacek Wroński, M.D., Ph.D, FEBU

Academic Editor

PLOS ONE

Dear Authors,

given the reviewer's opinion and the uniqueness of the topic, I accepted the revised version of the paper entitled "Contrast-Enhanced Micro-CT Imaging of a Foetal Female Pelvic Floor Reveals Anatomical Details" for publication in PLOS ONE.

With compliments

S. Wroński

Academic Editor

Reviewers' comments:

Reviewer's Responses to Questions

**Comments to the Author**

1. If the authors have adequately addressed your comments raised in a previous round of review and you feel that this manuscript is now acceptable for publication, you may indicate that here to bypass the “Comments to the Author” section, enter your conflict of interest statement in the “Confidential to Editor” section, and submit your "Accept" recommendation.

Reviewer #1: All comments have been addressed

2. Is the manuscript technically sound, and do the data support the conclusions?

Reviewer #1: Yes

3. Has the statistical analysis been performed appropriately and rigorously? 

Reviewer #1: N/A

4. Have the authors made all data underlying the findings in their manuscript fully available?

Reviewer #1: Yes

5. Is the manuscript presented in an intelligible fashion and written in standard English?

Reviewer #1: Yes

6. Review Comments to the Author

Reviewer #1: The authors have addressed all the reviewer comments, made the necessary adjustments where required. Their revisions have effectively clarified any concerns, strengthened the arguments, and ensured the overall quality of the manuscript. At this stage, no further revisions are needed.

7. PLOS authors have the option to publish the peer review history of their article (what does this mean? ). If published, this will include your full peer review and any attached files.

**Do you want your identity to be public for this peer review?** For information about this choice, including consent withdrawal, please see our Privacy Policy .

Reviewer #1: No

---

## [Editor Report · Acceptance letter]

PONE-D-24-49109R1

PLOS ONE

Dear Dr. Aaldijk,

I'm pleased to inform you that your manuscript has been deemed suitable for publication in PLOS ONE. Congratulations! Your manuscript is now being handed over to our production team.

Kind regards,

on behalf of

Dr. Stanisław Jacek Wroński

Academic Editor

PLOS ONE